# Biological Activity of Optimized Codon Bovine Type III Interferon Expressed in *Pichia pastoris*

**DOI:** 10.3390/v15051101

**Published:** 2023-04-30

**Authors:** Ran An, Runxiang Zhang, Yongli Guo, Jinfeng Geng, Minglu Si, Shuangfeng Wang, Mingchun Gao, Junwei Wang

**Affiliations:** 1Heilongjiang Provincial Key Laboratory of Zoonosis, College of Veterinary Medicine, Northeast Agricultural University, Harbin 150030, China; 2College of Animal Science and Technology, Northeast Agricultural University, Harbin 150030, China; 3Heilongjiang Provincial Key Laboratory for Infection and Immunity, Department of Immunology, Harbin Medical University, Harbin 150030, China

**Keywords:** bovine interferon–λ3, *Pichia pastoris*, codon optimization, glycosylation, antiviral activity, antiproliferative activity

## Abstract

Type III interferons (IFN–λs) exhibit potent antiviral activity and immunomodulatory effects in specific cells. Nucleotide fragments of the bovine *ifn–λ* (*boifn–λ*) gene were synthetized after codon optimization. The *boifn–λ* gene was then amplified by splicing using overlap extension PCR (SOE PCR), resulting in the serendipitous acquisition of the mutated boIFN–λ3^V18M^. The recombinant plasmid pPICZαA–boIFN–λ3/λ3^V18M^ was constructed, and the corresponding proteins were expressed in *Pichia pastoris* with a high–level extracellular soluble form. Dominant expression strains of boIFN–λ3/λ3^V18M^ were selected by Western blot and ELISA and cultured on a large scale, and the recombinant proteins purified by ammonium sulfate precipitation and ion exchange chromatography yielded 1.5g/L and 0.3 g/L, with 85% and 92% purity, respectively. The antiviral activity of boIFN–λ3/λ3^V18M^ exceeded 10^6^ U/mg, and they were neutralized with IFN–λ3 polyclonal antibodies, were susceptible to trypsin, and retained stability within defined pH and temperature ranges. Furthermore, boIFN–λ3/λ3^V18M^ exerted antiproliferative effects on MDBK cells without cytotoxicity at 10^4^ U/mL. Overall, boIFN–λ3 and boIFN–λ3^V18M^ did not differ substantially in biological activity, except for reduced glycosylation of the latter. The development of boIFN–λ3 and comparative evaluation with the mutant provide theoretical insights into the antiviral mechanisms of boIFN–λs and provide material for therapeutic development.

## 1. Introduction

As early as 2010, IFN–λ3 cDNA prediction sequences of bovine type III IFN were published in NCBI, and it was first cloned in 2011 [1]. It is located on chromosome 18 and contains 588 bases, encoding 195 amino acids and a signal peptide containing 23 amino acids. IFN–λ, which is also known as IL–28B, has remarkable antiviral activity and an immunomodulatory function and is expressed to varying degrees in different tissues and organs [2]. Human IFN–λ1 is a glycosylated protein, while IFN–λ2 and IFN–λ3 are not. The two members of the mouse type III interferon family (IFN–λ2 and IFN–λ3) are located on chromosome 7A3 of mice, with 98% similarity in their genes. It is worth noting that IFN–λ1 is a pseudogene, while both IFN–λ2 and IFN–λ3 undergo glycosylation modifications [3]. Antigen–presenting cells, such as dendritic cells, are the central cells that produce IFN–λ induced by a virus or RNA. The *Ifn–λ* gene is significantly expressed after infection with influenza virus or/and stimulation with pathogen–associated molecular patterns, such as lipopolysaccharide or poly I: C [4,5]. In skin tissue, IFN–λ is secreted by dendritic cells and regulatory T cells and has biological effects on keratinocytes and melanocytes [6]. IFN–λ3 proteins from humans and mice have been well characterized, but research on bovine IFN–λ3 is limited. To address viral infections in bovine, species–specific IFN–λ3 proteins are needed as early therapeutic agents to control viral infections. boIFN–λ3 has shown antiviral activity against foot–and–mouth disease virus (FMDV) and vesicular stomatitis virus in bovine cell cultures. Inoculation of cattle with Ad5–boIFN–λ3 induced systemic antiviral activity and up–regulation of IFN–stimulated gene expression in multiple tissues susceptible to FMDV infection. This makes boIFN–λ3 a potential biopharmaceutical candidate for further development to inhibit FMDV or other viruses in bovine [7].

*Pichia pastoris* combines the advantages of both prokaryotic and eukaryotic expression systems, with easy preparation and purification, is suitable for high–level expression, and is characterized not only by intracellular and secretory expression but also by post–translational modifications [8,9]. More than 500 proteins have been successfully expressed in the *P. pastoris* expression system [10]. Bovine IFN–α has been efficiently expressed through codon optimization of *P. pastoris*, whose expression yield is about three times that of the wild type at the same copies, and opti–boIFN–α hs shown antiviral activity in MDBK and IBRS–2 cells over 10^5^ U/mg against VSV [8]. *Pichia pastoris* expression involves modifying precursor proteins via proper folding, disulfide bond formation, and moderate O–glycosylation and N–glycosylation. Protein folding and disulfide bond formation are recognized as rate–limiting steps in the expression of exogenous proteins in *P. pastoris*. The organism’s ability to process, fold, and secrete recombinant proteins determines the productivity of the yeast expression system [11]. Over–glycosylation of yeast or protein in some natural active states is non–glycosylated, while yeast expression shows glycosylation, which is an unfavorable factor for *P. pastoris* expression. This type of protein often results in defects in biological activity and an increase in adverse reactions. Genetic engineering and exogenous gene modification technology can be used to mutate or modify glycosylation sites [8].

To achieve high–level expression of bovine IFN–λ3 (boIFN–λ3), we optimized boIFN–λ3, and achieved extracellular expression of optimized (opti)–boIFN–λ3 in *P. pastoris*. We amplified the 18th amino acid mutant of boIFN–λ3 by accident. This study intended to express boIFN–λ3/λ3^V18M^ by the *P. pastoris* production system. We compared the biological activity of the optimized codon between boIFN–λ3 and boIFN–λ3^V18M^. To further understand the biological function of boIFN–λ3 and provide materials for the preparation of antiviral agents.

## 2. Materials and Methods

### 2.1. Plasmid, Strains, Cells, Virus, and Antibody

The competent cell DH5α was preserved and used in our laboratory. The host strain GS115 and the expression vector pPICZαA were obtained from Invitrogen (Carlsbad, CA, USA). Madin–Darby bovine kidney (MDBK) cells were preserved in our laboratory. VSV (Vesicular Stomatitis Virus) was purchased from the China Institute of Veterinary Drug Control. Rabbit polyclonal antibodies (PAb) against human IL–28B (IFN–λ3) (GTX123335) were purchased from GeneTex (Irvine, CA, USA), and rabbit PAb against bovine IL–28B (IFN–λ3) was prepared by our laboratory.

### 2.2. Design and Synthesis of boIFN–λ3

Codon usage of predicting and cloned boIFN–λ3 (GenBank Accession No. XM_002695050 and No. HQ317919) from Bos taurus was analyzed using Dnastar Lasergene software and optimized by replacing a base frequently used in *P. pastoris* with frequently used codons and changed single–nucleotides using PCR amplification resulting in the 18th amino acid of the mature peptide, the former (called opti–boIFN–λ3, GenBank Accession No. OQ565418) being valine (V) and the latter (called boIFN–λ3V18M, GenBank Accession No. OQ565419) methionine (M). Opti–boIFN–λ3, referring to the principle of gene splicing by overlap extension (SOE), designed 14 oligonucleotide chains with about 20 bp overlapping each other (Table 1). BGI Co., Ltd. synthesized the oligonucleotide chains. The goal of the design was to minimize the free energy and secondary structure of boIFN–λ3 mRNA and obtain a nucleotide sequence that was theoretically most suitable for yeast expression.

### 2.3. Construction of a Recombinant Expression Vector

The *boifn–λ3* gene was amplified with SOE PCR, and each of the four oligonucleotide chains, P1~P4, P5~P8, P7~P10, and P11~P14, were fused to obtain S1~S4 product fragments, and then S1 and S2 products and S3 and S4 products were fused to obtain S12 and S34 fusion fragments, respectively. The fusion fragments were fused to obtain the template for boIFN–λ3 amplification, and P1 and P14 were then used for ordinary PCR amplification (Figure 1). The *boifn–λ3/λ3^V18M^* gene fragment was ligated to the pPICZαA vector between *Xho* I and *Xba* I sites, and the restriction enzyme (TaKaRa, Japan) sites were underlined corner markers. P2 was for cloning boIFN–λ3, while P2′ was for boIFN–λ3^V18M^. Then, recombinant plasmid pPICAαA–boIFN–λ3/λ3^V18M^ was obtained and transformed into *E. coli* DH5α (TIANGEN, Beijing, China). Positive transformants were identified using restriction analysis and sequencing. Then, the plasmids pPICZαA–boIFN–λ3/λ3^V18M^ and pPICZαA were linearized with *Sac* I and transformed into competent *P. pastoris* GS115 cells mediated separately by electroporation. After incubation in YPD (Yeast Extract–Peptone Dextrose) medium with zeocin at 30 °C for at least 72 h, the gene integration was verified by PCR using yeast genomic DNA as a template, and 5′AOX1(5′–GACTGGTTCCAATTGAGAAGC–3′) and 3′AOX1(5′–GCAAATGGCATTCTGACATCC–3′) as primers, and positive transformants were selected for expression.

### 2.4. Expression and Identification of boIFN–λ3/λ3^V18M^

The preparation method of *Pichia pastoris*–induced expression medium is as follows:(1)1 M potassium phosphate buffer pH6.0: 132 mL 1 M K_2_HPO_4_, 868 mL 1 M KH_2_PO_4_, filtration sterilization.(2)10× YNB (13.4% yeast nitrogen source alkali containing ammonium sulfate does not contain amino acids): Dissolve 134 g YNB in 1 L water, filter sterilization.(3)500× B biotin (0.02%): Dissolve 20 mg biotin in 100 mL water, filter sterilization.(4)BMGY (Buffered Glycerol–complex Medium) 1 L (Yeast growth medium): 1% yeast extract, 2% tryptone, 100 mM potassium phosphate pH6.0, 1.34% YNB, 5% biotin, 1% glycerol.(5)BMMY (Buffered Methanol–complex Medium) 1 L (yeast induction medium): 1% yeast extract, 2% tryptone, 100 mM potassium phosphate pH6.0, 1.34% YNB, 5% biotin, 0.5% methanol.

For example, for 1 L medium, 10 g yeast extract and 20 g tryptone were dissolved in 790 mL water and sterilized at 116 °C for 30 min, and then, 100 mL of 1 M potassium phosphate buffer, 100 mL 10× YNB, 2 mL 500× B biotin, and 10 mL 100% glycerol were added to prepare BMGY (5 mL methanol is added when preparing BMMY). It was used within 1 month of preparation.

The selected positive transformants transformed in *P. pastoris* GS115 cells were grown in BMGY at 28 °C until an OD600 of between 2 and 6 was reached and then centrifuged and resuspended with BMMY. After induction, the entire culture supernatant was harvested by centrifuging at 3000× *g* at 4 °C for 15 min. Then SDS–PAGE was performed to analyze the expression of boIFN–λ3/λ3^V18M^, and a Western blot was used to analyze the specificity of boIFN–λ3/λ3^V18M^ with rabbit PAb against human/bovine IL–28B.

### 2.5. Screening Dominant Expression Strains of boIFN–λ3/λ3^V18M^

Western blot analysis was performed to screen for protein expression. The supernatant was collected from the expressing strain after 72 h, and 6 μL was spotted onto a nitrocellulose membrane and allowed to dry naturally. PBST and the supernatant from induced GS115–pPICZαA were used as controls. The membrane was blocked with 5% skimmed milk PBST and incubated with a dilution of rabbit anti–human IL–28B primary antibody, followed by the HRP–labeled goat anti–rabbit secondary antibody. The membrane was washed and developed using a CN/DAB substrate kit (Thermo Scientific, Waltham, MA, USA) [12].

ELISA analysis was performed to screen for protein expression. The supernatant was collected from the expressing strain after 72 h and diluted appropriately with 0.05 M carbonate buffer (pH9.6) to serve as the coating solution. The solution was added to the ELISA plates and incubated at 4 °C overnight. The plates were then blocked with 5% skimmed milk and incubated with serial dilutions of rabbit PAb against the human IL–28B antibody. HRP–conjugated goat anti–rabbit IgG was used as the secondary antibody. Reactivity was visualized by color development using a chromogen/substrate mixture of 3,3′,5,5′–Tetramethylbenzidine/H_2_O_2_ (Sigma–Aldrich, Saint Louis, MO, USA). The reaction was terminated using 1 M H_2_SO_4_, and the absorbance of each well at 450 nm was measured using a microplate reader (Molecular Devices, Sunnyvale, CA, USA) [13].

### 2.6. Optimization of Expression Conditions

The selected recombinant strains integrated with the *boifn–λ3/λ3^V18M^* gene were cultivated in 25 mL of BMGY medium at 28 °C with constant shaking at 200× *g* until an OD600 of about 2. The recombinant protein was induced in a BMMY medium with methanol at a 0.5% final concentration. At 0 h, 6 h, 12 h, 24 h, 36 h, 48 h, 60 h, 72 h, 84 h, and 96 h, the supernatant was collected by centrifugation at 12,000× *g* for 30 min at 4 °C. Similarly, methanol was added to a final concentration of 0.5%, 1%, 1.5%, 2%, 3%, and 4% to maintain induction. Clarified supernatants were collected for SDS–PAGE analysis or protein purification.

### 2.7. Purification of boIFN–λ3

The expressed boIFN–λ3/λ3^V18M^ was purified with ammonium sulfate precipitation and ion exchange chromatography, and dialyzed with PBS for 48 h. The final protein concentration was determined using a BCA protein assay kit (Beyotime, Beijing, China). A detailed description is as follows.

For the ammonium sulfate precipitation method, samples of the supernatant of cleared yeast were poured into the beakers. While stirring at 4 °C with a final concentration of 60% ammonium sulfate. After the addition was complete, samples were left to stand overnight at 4 °C and then centrifuged at 5000× *g* for 15 min. Next, the supernatants were carefully removed, and pellets were resuspended in 50 mM Tris–HCl buffer, pH8.0, containing 0.15 M NaCl and 1 mM CaCl_2_, and dialyzed for 24 h to 48 h to remove NaCl. The suspensions were centrifuged at 12,000× *g* for 15 min to remove any remaining debris. The supernatants and dissolved pellets were analyzed for the presence of boIFN–λ3. For ion exchange and gel filtration chromatography, the ammonium sulfate fraction was applied to a column (1.25 × 6 cm) of Q–Sepharose FF (Pharmacia Biotechnology) equilibrated with 20 mM phosphate buffer, pH6.8. The column was washed with the same buffer, and the bound protein was eluted by the stepwise addition of buffers containing 0.25 M, 0.5 M, and 1.0 M NaCl. Fractions of 6 mL were collected and dialyzed again by PBS, and the protein concentration was determined after PEG20000 concentration. The collected solution was purified using Sephadex G–50 Sephadex (Pharmacia Biotechnology) gel filtration chromatography. Protein equilibrium and elution were performed with PBS.

### 2.8. Recombinant Protein Glycosylation Analysis

Purified recombinant boIFN–λ3 and boIFN–λ3^V18M^ were stained for glycoprotein analysis using the Protein Stains R Glycoprotein Gel Staining Kit from Sangon Biotech (Shanghai, China).

### 2.9. Antiviral Activity Assay

The antiviral activity of the recombinant boIFN–λ3/λ3^V18M^ titrated as described was measured on the MDBK/VSV system with modifications [14]. MDBK cells in 96–well plates were grown to approximately 90% confluence. Then, 100 μL of purified boIFN supernatant was diluted fourfold and in a serial dilution for 24 h and challenged with 100TCID50 VSV for another 24~48 h. Antiviral activity units (U) of IFN were calculated using the Reed–Muench method. The wells without viruses were considered mock–treated controls, and the wells without IFNs were designated as virus controls. One unit of interferon activity was defined as the amount required to inhibit the destruction of the cell monolayer by 50% [15]. Rabbit anti–bovine and rabbit anti–human IFN–λ3 PAb were diluted in a gradient and mixed with 100 U/mg/boIFN–λ3. Each PAb was diluted in a 96–well plate, and the inhibition effect of PAb on IFN was determined at 37 °C for 1 h. Rabbit anti–bovine IFN–α PAb was used as a negative control [16].

### 2.10. Physicochemical Characteristics Analysis of boIFN–λ3/λ3^V18M^

boIFN–λ3/λ3^V18M^ was treated with trypsin at a final concentration of 0.25% and then incubated at 37 °C for 1 h. boIFN–λ3/λ3^V18M^ was centrifuged at 15,000× *g* 4 °C for 20 min. The boIFN–λ3/λ3^V18M^ samples adjusted to pH2.0, 4.0, 10.0, and 12.0 with HCl and NaOH were incubated at 4 °C for 24 h, and then the pH was adjusted back to the original pH of boIFN–λ3/λ3^V18M^. The boIFN–λ3/λ3^V18M^ samples were separately incubated at 42 °C, 56 °C, and 63 °C for 2 h. The MDBK/VSV system was used to assess the antiviral activity with respect to that of the untreated boIFN–λ3/λ3^V18M^ samples [9,15,17].

### 2.11. Antiproliferative Assays of boIFN–λ3

The antiproliferative activity of boIFN–λ3 on MDBK cells was measured using the MTT assay [9]. Cells seeded in 96–well plates were treated with 0 U/mL, 10 U/mL, 100 U/mL, 1000 U/mL, and 10,000 U/mL boIFN–λ separately and then incubated with MTT for 4 h. DMSO was added after removing the culture medium, and OD490 was measured.

### 2.12. Statistical

All experiments were repeated three times independently. Data are expressed as the mean values with the standard deviation (SD). Statistical analyses were performed using GraphPad Prism software. The statistical significance of the differences between groups was determined by Student’s *t*–test and one–way analysis of variance (ANOVA) tests: * *p* < 0.05 and ** *p* < 0.01.

## 3. Results

### 3.1. Design of pPICZαA–boIFNλ3/λ3^V18M^ Plasmid

To improve the expression of boIFN–λ3 in *P. pastoris*, opti–*boifn–λ3* gene codons were adapted to the host genome’s preferred use, maintaining the amino acid sequence (Figure 2). The transcription of the gene contained five exons. The CDS was 588 bp, and the IFN precursor included a signal peptide of 19 amino acids. The mature peptide had 176 amino acids (Figure 2A). The mature peptide of the mammalian IFN–λ3 sequences contained six conserved cysteines that formed three disulfide bonds that stabilized the IFN–λ3 structure (Figure 2B). IFN–λ3 from various species consists of seven α–helices and five η–helices, with similar signal peptide cleavage sites. Comparison of boIFN–λ3 with other IFN–λ3 species showed a conservative amino acid similarity of 56.8~67.7%. After optimization, the Codon Adaptation Index increased to 0.92, with a GC content change from 64.72% to 45.80%, resulting in a total codon distribution exceeding 44% (Figure 2C,D).

### 3.2. Construction and Identification of pPICZαA–boIFNλ3/λ3^V18M^ Plasmid

Mature boIFN–λ3 was amplified at 545 bp (Figure 3A,B). pPICZαA–boIFN–λ3/λ3^V18M^ were amplified with 545 bp and 870 bp by the specific primer pair P1/P14 and 5′ AOX/P1, respectively. The recombinant gene fragment of 4037 bp was obtained by *Xho* I digestion, and two bands of 3506 bp and 531 bp were obtained by *Xho* I and *Xba* I digestion. The results suggested that enzyme digestion was consistent with the former and the sequencing results, which were successfully constructed (Figure 3C). A selected positive transformant was induced for expression.

### 3.3. High–Level Expression of a Synthetic boIFN–λ3 gene in P. pastoris

The pPICZαA–boIFN–λ3 plasmid was transformed into *P. pastoris* GS115 competent cells, yielding approximately 200 colonies on YPD plates (Appendix A). Thirteen white colonies exhibiting rapid growth, moderate size, smooth surfaces, moist protrusions, and neat edges were selected for further analysis. Eight positive recombinant strains were identified through amplification using primers 5′ AOX/3′ AOX, 5′ AOX/P14, P1/3′ AOX, and P1/P14. Six strains were confirmed after two rounds of PCR verification and selected for subsequent expression analysis. Induced expression of the recombinant plasmid after identification by specific primer PCR. Supernatants of pPICZαA–boIFN–λ3 clarified by centrifugation showed a visible boIFN–λ3 band identified using Western blot (Figure 4A) and ELISA (Figure 4B). The former was more intuitive and cumbersome; nevertheless, the latter were more sensitive. As a consequence, the ELISA method was used to screen high expression strains in later identification. Transformants with the boIFN–λ3 sequences showed high–level extracellular soluble optimized boIFN–λ3 expression with an estimated mass of 25 kDa. Using the same methodology, four strains of pPICZαA–boIFN–λ3^V18M^ were also selected for expression experiments (data not shown).

### 3.4. Optimized Expression of the Recombinant boIFN–λ3/λ3^V18M^

There were prominent expression bands at 23 kDa and 18 kDa, and the expression level gradually increased after 12 h and reached its peak after 72 h. The proportion of the total protein analyzed by the gray scale showed that the target protein accounted for more than 85% of the total protein. The target strip of 23 kDa accounted for about 70%, while the 18 kDa strip accounted for about 15%. Therefore, recombinant boIFN–λ3 and boIFN–λ3^V18M^ were highly expressed in *P. pastoris* and appeared as a glycosylated protein (23 kDa) and non–glycosylated protein (18 kDa) (Figure 5A,C). To compare the expression efficiency of boIFN–λ3 and boIFN–λ3^V18M^, we performed gray scale analysis of the target protein expression. The weighing method was used to measure the wet weight of the cell, and the expression curve of recombinant protein and the growth curve of yeast were obtained (Figure 5B,D). After 72 h, the expression no longer increased, while the expression of the miscellaneous protein increased. Therefore, the optimal induction time was 72 h in the baffled flasks. Both recombinant boIFN–λ3 and boIFN–λ3^V18M^ were expressed at 1.2 g/L, and there was no significant difference in expression between the two.

With the increase in the methanol concentration, the expression level of the target protein increased relatively and reached the peak at a 1.5% methanol concentration, while the expression level of heteroprotein also increased seriously. A 4% methanol concentration seriously inhibited the foreign protein, especially the recombinant boIFN–λ3 yeast strain, which tended toward over–glycosylation when the methanol concentration increased. These findings suggest that the proteins are mainly expressed in the form of glycosylation (Figure 5E,F).

### 3.5. Expression and Purification Analysis of the Recombinant boIFN–λ3/λ3^V18M^

SDS–PAGE and Western blot analysis revealed that the glycosylated and non–glycosylated proteins of recombinant boIFN–λ3 (Figure 6A–C) and boIFN–λ3^V18M^ (Figure 6D–F) were mainly expressed as soluble expression, which had good reactivity with anti–human IFN–λ3 and anti–bovine IFN–λ3 antibodies, indicating that boIFN–λ3 antigen had certain species conservation. The variation of its amino acid 18th of mature peptide did not affect its antigenicity.

BoIFN–λ3/λ3^V18M^ were purified by ammonium sulfate precipitation and ion exchange chromatography, respectively, both of which could reach a purity of over 80%, with the latter reaching a purity of over 90%. However, the purified protein expression of the latter was much lower than that of the former. Furthermore, there was no difference in the amount purified by boIFN–λ3 and boIFN–λ3^V18M^ using the same purification method (Table 2).

### 3.6. Glycosylation Analysis of the Recombinant boIFN–λ3/λ3^V18M^

After dialysis, the recombinant boIFN–λ3 protein was subjected to cation exchange chromatography (Figure 6G). The glycoprotein of 23 kDa that was dyed red and non–glycosylated at 18 kDa was not stained. It proved that most boIFN–λ3 was expressed in the form of glycoprotein (Figure 6H). Prediction of boIFN–λ3^V18M^ was consistent with that of boIFN–λ3, so the change of amino acid 18 of the mature peptide did not change its glycosylation site (Appendix A).

### 3.7. Antiviral Activities of the Recombinant boIFN–λ3/λ3^V18M^

The results showed that the recombinant protein effectively inhibited the pathogenicity of VSV to MDBK cells (Figure 7A). The antiviral activities of boIFN–λ3 and boIFN–λ3^V18M^ were >10^6^ U/mg. boIFN–λ3 by ammonium sulfate precipitation was 2.14 × 10^6^ U/mg, and by ion–exchange chromatography, it was 2.39 × 10^6^ U/mg. boIFN–λ3^V18M^ by ammonium sulfate precipitation was 1.93 × 10^6^ U/mg, and by ion–exchange chromatography, it was 2.15 × 10^6^ U/mg. The difference in antiviral activity between boIFN–λ3 and boIFN–λ3^V18M^ was not significant, so the difference in amino acid 18 of the mature peptide did not affect the biological activity of boIFN–λ3 on MDBK–VSV.

### 3.8. Antibody Neutralization for the Recombinant boIFN–λ3/λ3^V18M^

After boIFN–λ3 and boIFN–λ3^V18M^ were incubated with rabbit anti–bovine and rabbit anti–human IFN–λ3 PAb at 37 °C for 1 h, the antiviral activity of 100 U/mL IFN–λ3 could be completely neutralized by the high concentration of the antibody, while rabbit anti–bovine IFN–α pAb could not reduce the antiviral activity of IFN–λ3 (Figure 7B,C).

### 3.9. Physicochemical Characteristic of boIFN–λ3/λ3^V18M^

After treatment with 0.25% trypsin, the antiviral activity of boIFN–λ3/λ3^V18M^ entirely disappeared, suggesting that they are highly sensitive to trypsin. boIFN–λ3 was centrifuged at 15,000× *g* at 4 °C for 20 min. Its biological activity on MDBK/VSV was unchanged, and its titer was 2.32 × 10^6^ U/mg, indicating existed in soluble form.

The lower degree of decrease in protein activity at different temperatures indicates that recombinant boIFN–λ3/λ3^V18M^ is less sensitive to heat, and the results are shown in Figure 8A.

The antiviral activity of boIFN–λ3 was slightly decreased by different pH, and it was more acid tolerant than alkali. The antiviral activity of the recombinant protein merely decreased more than two times at pH2 and even more decreased about 1.5 times at pH4 and 12 (Figure 8B). The physicochemical properties of boIFN–λ3^V18M^ were consistent with those of boIFN–λ3.

### 3.10. Antiproliferative of boIFN–λ3/λ3^V18M^

Microscopic observation showed that there was no difference between the cells treated with boIFN–λ3 and normal cells (Figure 9 and Appendix A). MTT assay showed that MDBK cells were treated with gradient–diluted purified boIFN–λ3 for 72 h. It was found that the cell activity of the experimental group was slightly decreased compared with that of the non–IFN–treated group. BoIFN–λ3 of 100 U/mL, 1000 U/mL and 10,000 U/mL on MDBK cells decreased by 0.89%, 2.45%, and 8.56%, respectively. It was considered that the dose used had no cytotoxicity, and the high dose of >10,000 U/mL had little effect on proliferation inhibition and cytotoxicity. Similarly, boIFN–λ3^V18M^ had almost no cytotoxicity, and 100 U/mL, 1000 U/mL and 10,000 U/mL of boIFN–λ3^V18M^ on MDBK cells decreased by 1.21%, 5.45%, and 8.25%, respectively.

## 4. Discussion

IFN–λs, as a type III interferon family, have potential development in the treatment of viral infection, as they have more excellent antiviral and immunomodulatory effects than type I IFNs [18]. To obtain a large number of bioactive boIFN–λ3 similar to native boIFN–λ3. We prepared the optimized *boifn–λ3* gene by considering the yeast codon usage preference, mRNA free energy, and the secondary structure [19], which was designed and developed. After optimization, the codon adaptation index was improved, and the GC content was decreased. However, the free energy after optimization was not lower than before; in other words, the structure was not stable after optimization, indicating that the natural conformation was the most suitable structure for preferential stability.

We optimized the methanol concentration and induction time, which play an essential role in the amount of protein secreted and expressed by *P. pastoris*. Methanol is the principal carbon source and gene expression inducer in *P. pastoris* fermentation strategy, and its level affects cell growth and productivity. The expression of recombinant protein requires at least 0.5% methanol concentration, with a methanol–induced expression of the optimal concentration of up to 2~2.5%. A low concentration of methanol will cause the proteolytic degradation of the expressed protein. In addition, methanol with a concentration greater than 5% is toxic to cells [20]. Our results showed that with the increase in the methanol concentration, the expression of the target protein also increased, and after a certain methanol concentration reached the peak, the expression of the protein was significantly inhibited under a high concentration of methanol, indicating that with the increase in the methanol concentration, the cell may be damaged, and its expression decreased. This is basically consistent with the above reviews.

Incubation time is one of the factors affecting the highest protein expression in the *P. pastoris* expression system. About 100 h of production time is relatively long. The incubation time was related to the number of yeast cells and the degree of target protein degradation. Studies have shown that *P. pastoris* cells grow fastest at 96 h, whereas protein expression is highest at 48 h. It is likely that a longer incubation time causes more proteolytic digestion of the expressed protein. Some studies suggest an optimal time for protein expression at 72 to 96 h [20]. Our results confirm this: we obtained an optimal expression time within 100 h and determined the optimal expression at 72 h. With the further extension of time, the expression level was not obvious and showed a downward trend, which may be caused by protein hydrolysis. In addition, in *AOX* gene subtypes, the sorbitol concentration and temperature factors affect the expression of the target protein, which is what we need to consider in the future to further optimize the expression.

In this study, *P. pastoris* secreted boIFN–λ3 and the target protein accounted for more than 85% of the total protein. Therefore, in the aspect of purification, the ammonium sulfate salting–out method was first selected for crude extraction. Then, depending on the theoretical isoelectric point of boIFN–λ3 for the characteristics of 8.31 and the molecular weight between 17~23 kDa, the crude protein was purified by cation exchange chromatography and gel filtration chromatography. Unfortunately, only the former purity was 92% of the target protein. Gel filtration chromatography did not further identify the target protein. Even if the gel medium was Sephadex G–50, the upper column’s volume and eluent were not obtained. It may pertain to the method of purification and the quality of gel. Even so, the purified recombinant protein was achieved by ammonium sulfate salting out and ion exchange chromatography. Recombinant proteins secreted by *P. pastoris* are released into the culture supernatant and are easily purified due to the limited production of endogenous secretory proteins [20]. This yeast system is convenient; it also has relatively rapid expression times and co–translational and post–translational processing, especially for large–scale industrialization.

For interferon therapeutic proteins to attain complete biological activity, glycosylation must play an important role. *P. pastoris* is capable of performing post–translational modifications, such as both N– and O–linked glycosylation [19]. From glycosylation site prediction and glycoprotein staining, we concluded that recombinant boIFN–λ3 was expressed in glycosylated and non–glycosylated forms, consistent with the literature reports [21]. SDS–PAGE and Western blot showed that it is mainly expressed in the glycosylated form. In the expression and purification of boIFN–λ3^V18M^, the glycosylation protein was reduced. The glycosylated boIFN–λ3 has good water solubility, and the purified boIFN–λ3 results in less non–glycosylation, better hydrophilicity, and higher yields. Although boIFN–λ3^V18M^, whose alteration site was not at the glycosylation site, was purified, the non–specific bands were decreased, and its level of glycosylation was higher, indicating better water solubility. This is consistent with our general consensus that N–linked glycosylation constitutes a pivotal role in its hydrodynamic volume in therapeutic proteins and therefore in its pharmacodynamics behavior, which further reflects that the change in the 18th amino acid affected its glycosylation level [19]. Moreover, very little O–linked glycosylation has been observed in *P. pastoris* [20]. This is inconsistent with the number of glycosylation site categories predicted by software (Putative: One N–gly; two O–gly). Interestingly, there was no difference in the good performance of the physicochemical properties and antiviral activity of boIFN–λ3/λ3^V18M^. In future production practice, boIFN–λ3/λ3^V18M^ can be mixed and used together, which can achieve both increased yield and increased solubility, as well as antiviral activity. To further explore the effect of glycosylation on boIFN–λ3, the subsequent work needs to use glycosidase decomposition to obtain a single protein for research.

Recombinant boIFN–λ3 is neutralized by specific antibodies, giving it the following physicochemical properties: heat resistant and acid and alkali resistant [9]. These properties are comparable to those of known type I interferons, confirming the functional similarity between IFN–λ and IFN–α [22]. In this study, the effect of the 18th amino acid (V–M) change on the expression and activity of recombinant boIFN–λ3 was analyzed and showed that the difference in amino acid 18 of the mature peptide did not affect the biological activity and physicochemical properties of recombinant boIFN–λ3. Gad’s research team found that each of the 16 amino acids exposed to IFN–λ3 at the helix A, D, F, and the corner of AB was mutated into alanine, which showed that the activity of the mutant was different after the mutation. The 158th amino acid was the critical amino acid for its binding receptor, and it was inactivated after mutation [23]. Therefore, the 18th amino acid of the mature peptide is a dispensable amino acid for IFN–λ3 to play its role, and the structural characteristics of boIFN–λ3 do not hinder its antiviral activity.

## 5. Conclusions

In conclusion, we successfully constructed the glycosylation–modified boIFN–λ3 and incidental boIFN–λ3^V18M^ with only the 18th amino acid changed from valine to methionine by optimizing the design of the boIFNλ3 sequence and by optimizing the expression conditions of *P. pastoris*. With the ammonium sulfate precipitation method, boIFN–λ3 and boIFN–λ3^V18M^ were 2.14 × 10^6^ U/mg and 1.93 × 10^6^ U/mg, respectively; by the ion exchange chromatography method, they were 2.39 × 10^6^ U/mg and 2.15 × 10^6^ U/mg, respectively. The expression, purification, physicochemical properties, and cell proliferation of the recombinant boIFN–λ3/λ3^V18M^ protein were consistent with the amino acid changes, which did not affect their characteristics. To further understand the biological function of boIFN–λs, a theoretical basis and material for the preparation of antiviral agents should be provided.

## Figures and Tables

**Figure 1 viruses-15-01101-f001:**
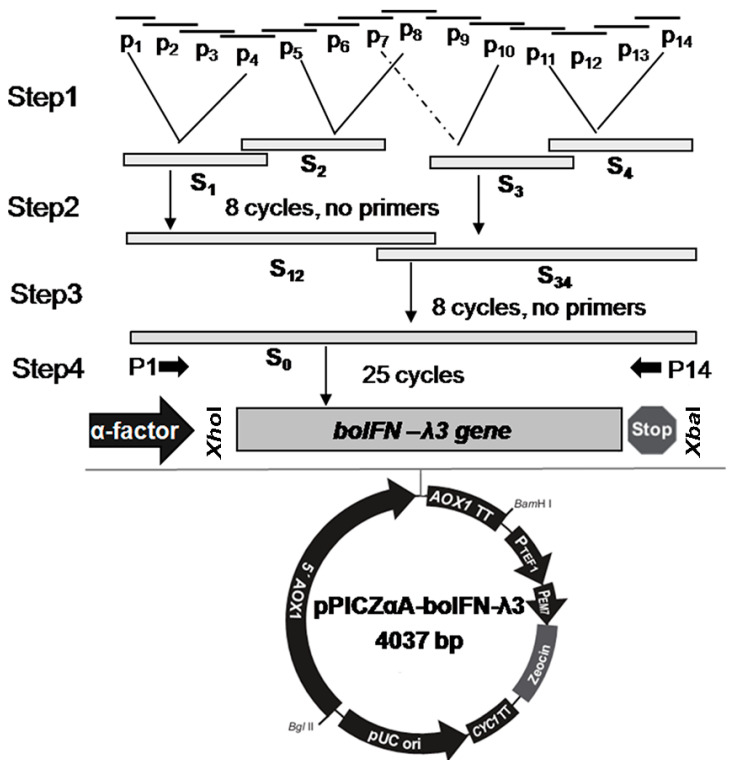
Construction of pPICZαA–boIFN–λ3 by SOE with optimized codons.

**Figure 2 viruses-15-01101-f002:**
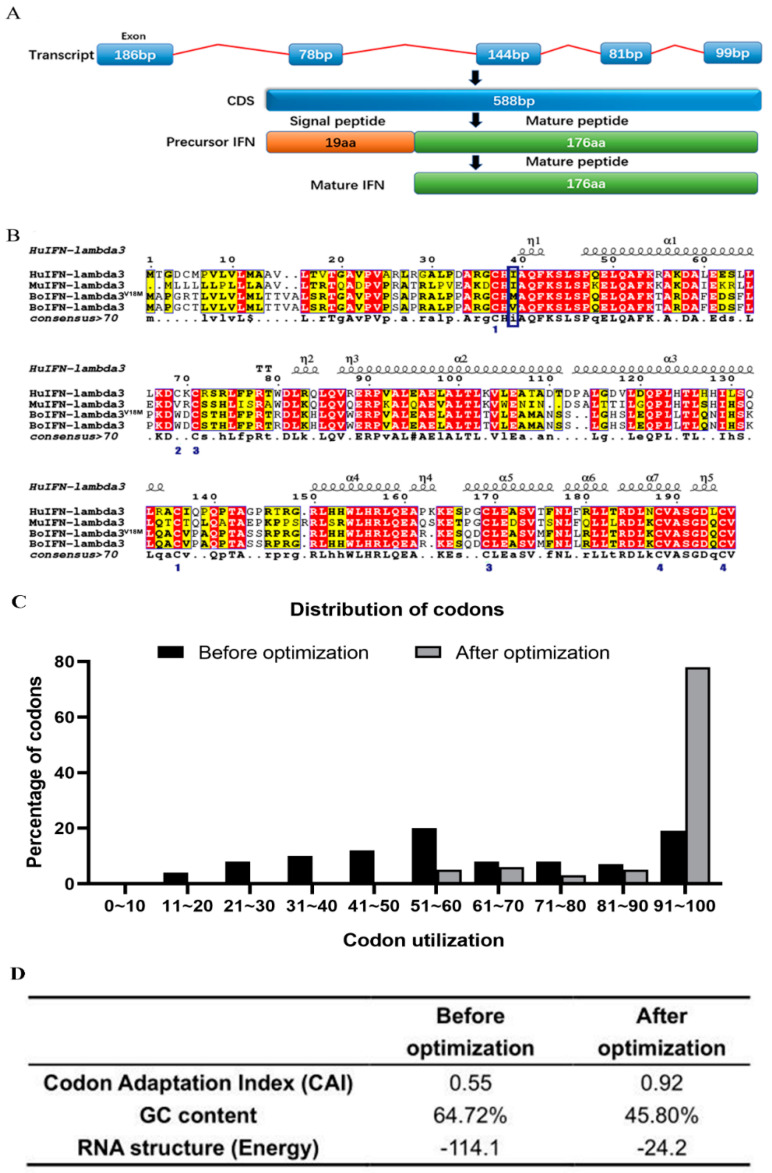
Sequence alignment of codon–optimized boIFN–λ3^V18M^ and original boIFN–λ3. (**A**). The translation process of IFN–λ3. (**B**). Sequence alignment of IFN–λ3. GenBank Accession: HuIFN–lambda3, NM_172139.2; MuIFN–lambda3, NM_177396.1; BoIFN–lambda3V18M, No. OQ565419; BoIFN–lambda3, No. OQ565418. Identical and similar residues are boxed in red and yellow, respectively. The residues representing the mutation site are within the indigo frame. (**C**). Optimization for codon usage rate of the bovine *ifn–λ3* gene in *P. pastoris*. The percentage distribution of codons was computed in codon quality groups. The value of 100 was set for the codon with the highest usage frequency for a given amino acid in the desired expression organism. Codons with values lower than 30 are likely to hamper expression efficiency. (**D**). Comparison of the values of index optimization. The possibility of a high protein expression level is correlated with the value of CAI (a CAI of 1.0 is considered to be ideal, whereas a CAI of >0.8 is rated as good for the desired expression). The ideal percentage range of GC content is between 30% and 70%. Any peaks outside this range will adversely affect transcriptional and translational efficiency.

**Figure 3 viruses-15-01101-f003:**
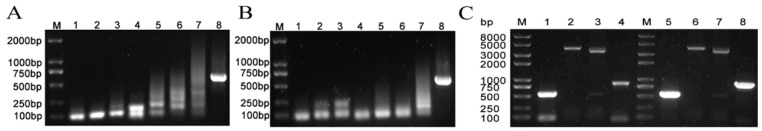
Identification of boIFN–λ3/λ3^V18M^. (**A**,**B**). boIFN–λ3 and boIFN–λ3^V18M^ SOE–PCR, respectively. M: Trans2K DNA Marker; 1~4, 5~6, 7, and 8 indicate the first, second, third, and fourth PCR results, respectively. (**C**). Enzyme digestion and PCR identification of the *P. pastoris* transformants integrated into the boIFN–λ3 gene. M. Trans2K PlusII DNA Marker; 1 and 5. PCR result based on P1/P14; 2 and 6. *Xho* I digested; 3 and 7 *Xho* I and *Xba* I digested; 4 and 8. PCR result based on 5′AOX/P14. 1~4 was the result of recombinant expression vector pPICZαA–boIFN–λ3 and 5~8 indicated the pPICZαA–boIFN–λ3^V18M^.

**Figure 4 viruses-15-01101-f004:**
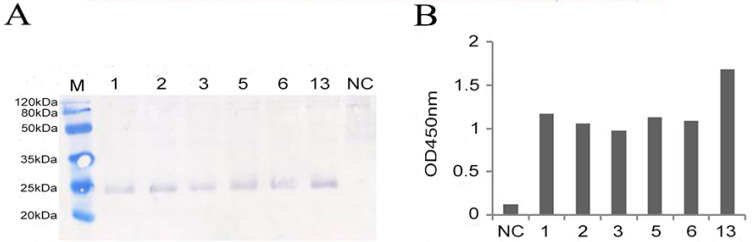
Selection of highly expressed recombinant GS115–pPICZαA–boIFN–λ3. The numbers 1, 2, 3, 5, 6, and 13 indicate recombinant GS115–pPICZαA–boIFN–λ3, and NC indicates GS115–pPICZαA. (**A**). Identification results of expressed recombinant GS115–pPICZαA–boIFN–λ3 by Western blot. (**B**). The results of supernatants of pPICZαA–boIFN–λ3 were identified by ELISA.

**Figure 5 viruses-15-01101-f005:**
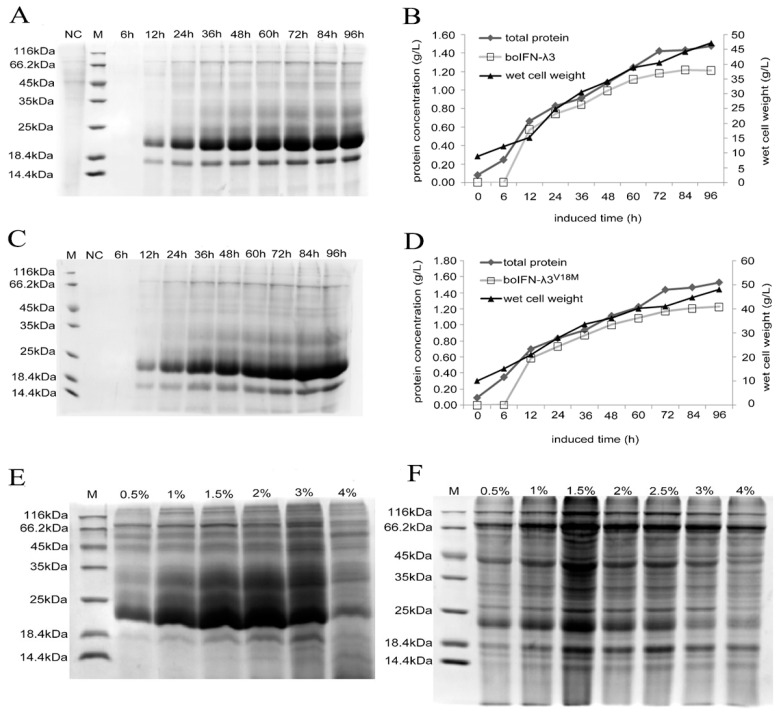
Optimization of the expression conditions of the recombinant boIFN–λ3/λ3^V18M^. (**A**,**C**) The SDS–PAGE analysis of different induction times of the recombinant boIFN–λ3 and boIFN–λ3^V18M^, respectively. (**B**,**D**). The protein weight analysis of different induction times of recombinant boIFN–λ3 and boIFN–λ3^V18M^, respectively. (**E**,**F**). SDS–PAGE analysis of recombinant *P. pastoris* GS115 under different methanol concentrations. (**E**). GS115–pPICZαA–boIFN–λ3. (**F**). GS115–pPICZαA–boIFN–λ3^V18M^. M. Unstained protein molecular weight Marker; NC. The induced protein of GS115–pPICZαA.

**Figure 6 viruses-15-01101-f006:**
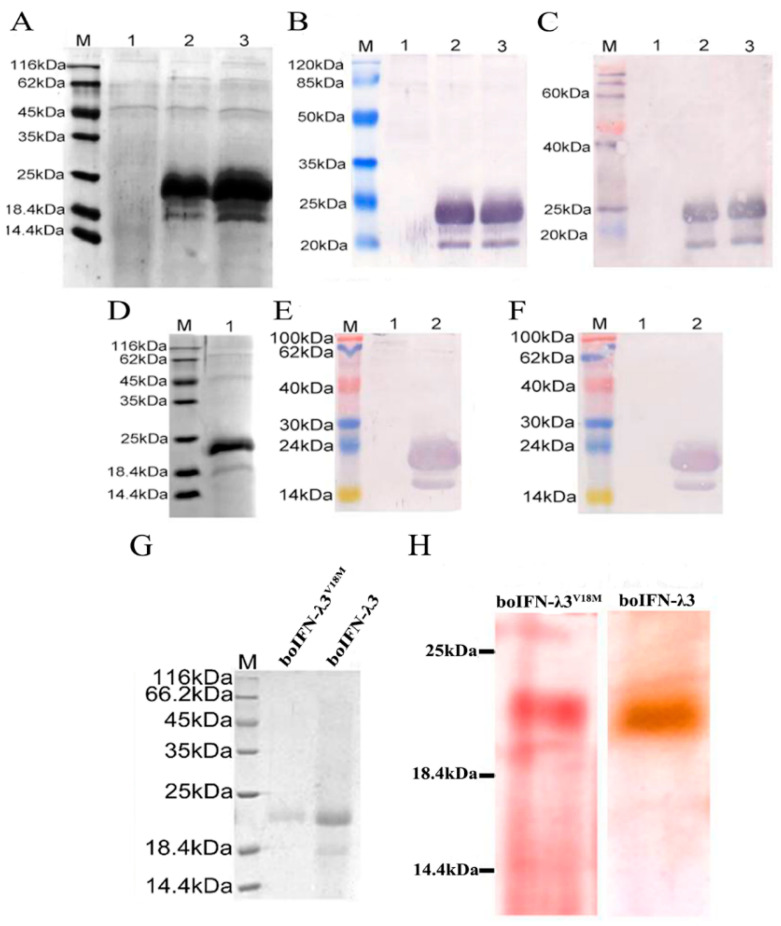
Protein identification and glycoprotein analysis of recombinant boIFN–λ3/λ3^V18M^. (**A**). SDS–PAGE for recombinant boIFN–λ3 strains 5 and 13. M. Unstained protein molecular weight Marker, 1. The induced protein of GS115–pPICZαA, 2 and 3 is the induced protein of GS115–pPICZαA–boIFN–λ3 strains 5 and 13, respectively. (**B**,**C**). Western blot analysis of rabbit anti bovine IFN–λ3 polyclonal antibody (**B**) and rabbit anti human IFN–λ3 polyclonal antibody (**C**) for immune response of recombinant boIFN–λ3. M. PageRuler Marker; 1. The induced protein of GS115–pPICZαA; 2,3. The induced protein of GS115–pPICZαA–boIFN–λ3 strains 5 and 13, respectively. (**D**). SDS–PAGE for recombinant boIFN–λ3^V18M^ strain 2. M. Unstained protein molecular weight Marker; 1. The induced protein of GS115–pPICZαA–boIFN–λ3^V18M^ strain 2. (**E**,**F**). Western blot analysis of rabbit anti bovine IFN–λ3 polyclonal antibody (**E**) and rabbit anti human IFN–λ3 polyclonal antibody (**F**) for immune response of recombinant boIFN–λ3^V18M^. M. PageRuler Marker; 1. The induced protein of GS115–pPICZαA; 2. The induced protein of GS115–pPICZαA–boIFN–λ3^V18M^ strain 2. (**G**). SDS–PAGE analysis of the purification of recombinant boIFN–λ3 and boIFN–λ3^V18M^. (**H**). Glycoprotein staining of recombinant boIFN–λ3 and boIFN–λ3^V18M^.

**Figure 7 viruses-15-01101-f007:**
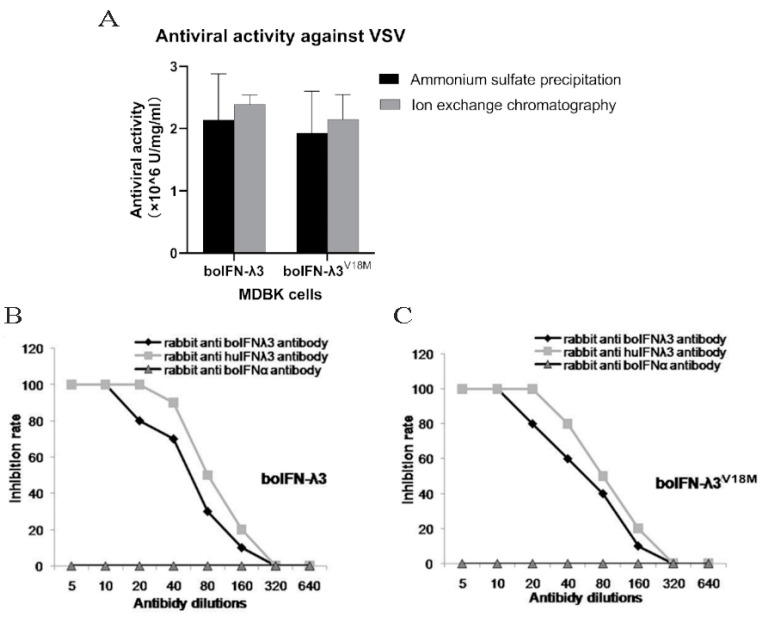
Antiviral activity analysis of the recombinant boIFN–λ3/λ3^V18M^ (**A**) and antibody neutralization test for the recombinant boIFN–λ3 (**B**) and boIFN–λ3^V18M^ (**C**).

**Figure 8 viruses-15-01101-f008:**
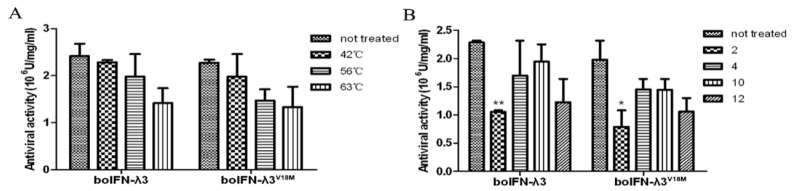
The characteristics of recombinant boIFN–λ3 and boIFN–λ3^V18M^. (**A**). The results of the thermal stability test of the recombinant protein. (**B**). Acid and alkali resistance of the recombinant protein. (* *p*–value < 0.05; ** *p*–value < 0.01).

**Figure 9 viruses-15-01101-f009:**
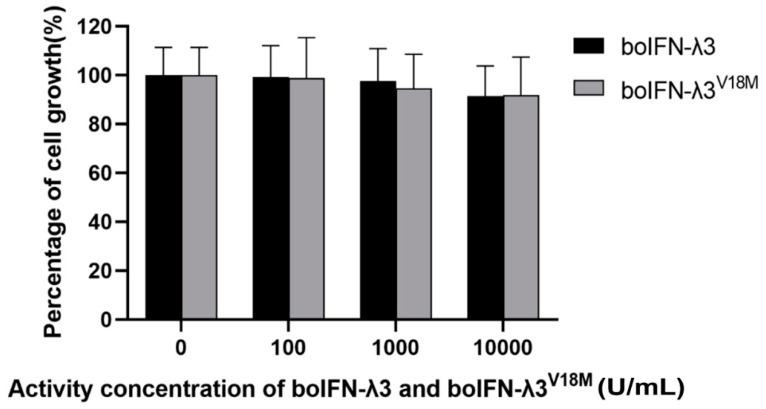
Antiproliferative analysis of boIFN–λ3/λ3^V18M^.

**Table 1 viruses-15-01101-t001:** Oligonucleotides for synthesizing the boIFN–λ3 gene.

No.	Oligonucleotide Fragments (5′–3′)
P1	AGCTCTCGAG_XhoI_AAAAGAGTTCCAGTTCCATCTGCCCCAAGAGCTTTGCC
P2	AGACAAGGACTTAAATTGAGCCACGTGACAACCACGGGCTGGTGGCAAAGCTCTTGGGG
P2′	AGACAAGGACTTAAATTGAGCCATGTGACAACCACGGGCTGGTGGCAAAGCTCTTGGGG
P3	GCTCAATTTAAGTCCTTGTCTCCACAAGAATTGCAAGCCTTTAAGACTGCTAGAGACGC
P4	AGAACAGTCCCAGTCCTTTGGCAAGAAAGAGTCTTCGAAAGCGTCTCTAGCAGTCTTAA
P5	AAGGACTGGGACTGTTCTACTCACTTGTTCCCAAGAACTAGAGACTTGAAGCACTTGCA
P6	AGGCCAATTCAGCTTCCAAAGCAACAGGTCTTTCCCAAACTTGCAAGTGCTTCAAGTCT
P7	GGAAGCTGAATTGGCCTTGACTTTGACTGTTTTGGAAGCTATGGCTAACTCTTCTTTGG
P8	GTTTTGCAAAGTCAACAATGGCTGTTCCAAAGAGTGACCCAAAGAAGAGTTAGCCATAG
P9	CATTGTTGACTTTGCAAAACATCCACTCTAAGTTGCAAGCCTGTGTTCCAGCTCAACCA
P10	CAACCAGTGGTGCAATCTACCTCTAGGTCTGGAAGAGGCGGTTGGTTGAGCTGGAACAC
P11	AGATTGCACCACTGGTTGCACAGATTGCAAGAGGCTAGAAAGGAATCCCAAGACTGTTT
P12	AGTCAACAATCTCAACAAGTTGAACATAACAGAAGCCTCCAAACAGTCTTGGGATTCCT
P13	AACTTGTTGAGATTGTTGACTAGAGACTTGAAGTGTGTTGCTTCTGGTGACCAATGTGT
P14	GACTTCTAGA*_Xba_*_I_TTAAACACATTGGTCACCAGAAG

**Table 2 viruses-15-01101-t002:** Purification of the recombinant boIFN–λ3 and boIFN–λ3^V18M^.

100 mL Volume		Total Protein (mg)	Target Protein (mg)	Purity (%)
BoIFN–λ3	Ammonium sulfate precipitation	180	152.1	84.5
	Ion exchange chromatography	32	29.44	92
BoIFN–λ3^V18M^	Ammonium sulfate precipitation	175	148.75	85
	Ion exchange chromatography	34	31.45	92.5

## Data Availability

The raw data supporting the conclusions of this article will be made available by the authors without undue reservation.

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
