# Peer review of "Biological Activity of Optimized Codon Bovine Type III Interferon Expressed in Pichia pastoris"

_viruses, 2023, doi:10.3390/v15051101_

Round 1

Reviewer 1 Report

This is a straight forward manuscript that describes the generation of bovine IFNl3.  Characterization of the recombinant protein is thorough and they demonstrate anti-viral activity but not cytotoxicity.  They also characterize a variant V18M and conclude that the amino acid change did have any significant biological impact.  I only have a few minor comments:

1. It would be of interest to know if the recombinant proteins have a biological activity that is comparable to recombinant IFNa but it is appreciated that an equivalent IFNa protein may not be available.

2. Have the proteins ever been injected into pigs in order to evaluate if they are immunogenic?  Given the data of anti-human IFNa antibodies in certain patients, this would be of interest.

3. The authors should speculate on the use of these interferons on porcine virus infections especially since oral IFNs have been reported to have biological activity in swine.

English is good

Reviewer 2 Report

The article titled "Biological activity of Optimized Codon bovine type III interferon expressed in Pichia pastoris" underwent review, where the authors optimized the gene of bovine interferon lambda-3 (boIFN-λ3) and generating two genes that encode a protein: boIFN-λ3 with the tentatively native sequence and boIFN-λ3V18M with a modification in amino acid 18 of the mature peptide by changing a valine to a methionine using the Pichia pastoris pPICZαA secretion expression system. The authors achieved a significant level of recombinant protein expression into the culture medium by optimizing the concentration of methanol addition and the duration of induction expression. The biophysical properties, cytotoxicity and antiviral activity of the proteins was also determined, and the results showed their potential for subsequent application.

However, there are still some issues that need to be addressed before publication.

Q1: Lines 35 state that "23 signal peptides", the description is inaccurate, it should be described as “a signal peptide containing 23 amino acids”.

Q2: The introduction provides a good overview of IFN-λ3 and its potential as an antiviral agent but does not sufficiently justify why boIFN-λ3 was chosen as a research subject or potential biopharmaceutical. The authors should build a stronger case for studying boIFN-λ3.

Q3: Lines 118, “3000 × g”, it should be “3,000 × g”, In the manuscript, insert a comma after every three digits for all numbers exceeding one hundred.

Q4: The materials and methods need more detail to enable reproducibility. The procedure for codon optimizing the genes, the media formulations used for protein expression, and details of the antiviral assays should be described fully.

Q5: The authors should boost the protein yields by further optimizing expression conditions before broader use or distribution. 1.5 g/L seems rather low.

Q6: Lines 468-469, the conclusions are somewhat overstated given the limited data. Claims that the proteins "theoretical basis" or "antiviral mechanisms" are premature without in vivo data or studies exploring the mechanisms of boIFN-λ3 action. The conclusions should be toned down until further research is conducted. Please reformulate your concluding remarks, or excise this sentence.

Q7: Checking for any accidental duplication of words or grammatical/spelling errors to polish the language.

Q8: Figure 4, 5 and 6. Labeling the western blot images with protein names/sizes for clarity.

Q9: The significance of this work largely depends on in vivo data that establish the therapeutic potential of boIFN-λ3. The authors may want to consider a follow-up publication focusing on the potential immunogenicity of boIFN-λ3 in cattle and how this could impact veterinary application of this biologic. In vivo efficacy against viral infection or disease models are recommended and species cross-reactivity and neutralizing antibody formation are important safety considerations.

Q10: This study could be furthered by constructing variants of boIFN-λ3 (including boIFN-λ3V18M) to investigate the relationship between structure and function, enhance efficacy, or create novel immunomodulatory forms. A collection of boIFN-λ3 variants with unique properties could represent a promising avenue for future research.

Overall, this manuscript presents the initial stages of a potentially valuable research line. The authors have engineered a system to produce a novel interferon lambda and shown preliminary indications of function. Addressing the points raised would convey the true potential significance of these results and enable this paper to reach its full impact.
